# Mechanistic Studies of Antibiotic Adjuvants Reducing Kidney’s Bacterial Loads upon Systemic Monotherapy

**DOI:** 10.3390/pharmaceutics13111947

**Published:** 2021-11-17

**Authors:** Fadia Zaknoon, Ohad Meir, Amram Mor

**Affiliations:** Department of Biotechnology and Food Engineering, Technion-Israel Institute of Technology, Haifa 3200003, Israel; fadia@technion.ac.il (F.Z.); mohad@campus.technion.ac.il (O.M.)

**Keywords:** antibiotic resistance, innate immunity, synergism of action, peptidomimetics, mechanism of action

## Abstract

We describe the design and attributes of a linear pentapeptide-like derivative (C_14(ω5)_OOc_10_O) screened for its ability to elicit bactericidal competences of plasma constituents against Gram-negative bacteria (GNB). In simpler culture media, the lipopeptide revealed high aptitudes to sensitize resilient GNB to hydrophobic and/or efflux-substrate antibiotics, whereas in their absence, C_14(ω5)_OOc_10_O only briefly delayed bacterial proliferation. Instead, at low micromolar concentrations, the lipopeptide has rapidly lowered bacterial proton and ATP levels, although significantly less than upon treatment with its bactericidal analog. Mechanistic studies support a two-step scenario providing a plausible explanation for the lipopeptide’s biological outcomes against GNB: initially, C_14(ω5)_OOc_10_O permeabilizes the outer membrane similarly to polymyxin B, albeit in a manner not necessitating as much LPS-binding affinity. Subsequently, C_14(ω5)_OOc_10_O would interact with the inner membrane gently yet intensively enough to restrain membrane-protein functions such as drug efflux and/or ATP generation, while averting the harsher inner membrane perturbations that mediate the fatal outcome associated with bactericidal peers. Preliminary in vivo studies where skin wound infections were introduced in mice, revealed a significant efficacy in affecting bacterial viability upon topical treatment with creams containing C_14(ω5)_OOc_10_O, whereas synergistic combination therapies were able to secure the pathogen’s eradication. Further, capitalizing on the finding that C_14(ω5)_OOc_10_O plasma-potentiating concentrations were attainable in mice blood at sub-maximal tolerated doses, we used a urinary tract infection model to acquire evidence for the lipopeptide’s systemic capacity to reduce the kidney’s bacterial loads. Collectively, the data establish the role of C_14(ω5)_OOc_10_O as a compelling antibacterial potentiator and suggest its drug-like potential.

## 1. Introduction

Towards tackling the ongoing antibiotic resistance crisis, the search for antibiotics potentiators is gaining increasing interest [1,2,3] as a backup alternative for development of brand new substitutes. Namely, broadening the activity spectrum of established antibiotics counts as a tempting approach for minimizing the emergence and impact of resistance, particularly when the antibiotics inefficacy against Gram-negative bacteria (GNB) emanates from low permeability across the outer membrane (OM) [4,5]. In this sense, antimicrobial peptides (AMPs) represent appealing potential substitutes [6,7,8] as their antibacterial properties largely depend on molecular hydrophobicity which, in turn, can be synthetically fine-tuned with relative ease. Indeed, unlike outright hydrophobic AMPs that tend to disrupt both membranes of GNB abruptly [9,10,11], borderline hydrophobic analogs were proposed to maintain the OM permeabilization capacity but may additionally instigate little more than transient superficial damages to the inner membrane (IM) [12,13,14,15]. While not fully understood, the latter activity was linked to a variety of processes (such as restricted efflux [16], inhibited expression of antibiotic resistance factors [17,18,19] and pathogens sensitization to diverse antimicrobials [20,21,22,23,24]) some of which, may open a window of opportunity for therapeutic exploitation. Thus, potentiated agents would encompass exogenous antibiotics as well as endogenous bactericidal capabilities of innate plasma complements [25].

From this perspective, the AMP mimetic approach based on oligomeric acylated cations (OAC) [26] appears particularly suitable for engineering membrane active selective compounds [10,27,28], as it provides a simple, sensitive, and systematic tool for dissecting the relative importance of two most critical AMP attributes, charge and hydrophobicity, as will be illustrated herein. Recent OAC designs [29,30,31,32,33] have concentrated on the pentameric formula **A_1_C_1_C_2_A_2_C_3_**, where As and Cs represent acyl derivatives and cationic amino acids, respectively. Among the sequences investigated so far, C_14_KKc_12_K (Figure 1a) revealed broad-spectrum bactericidal properties [32]. However, at least from therapeutic perspectives, this compound also exhibited caveat properties (such as aggregation in aqueous media) [32] that seem interconnected to its high hydrophobicity. In addition, as will be revealed in the present study, C_14_KKc_12_K holds potential for high hemolytic activity and for inactivity in plasma. Therefore, we set out to address these flaws by investigating a series of lower hydrophobicity analogs working under the hypothesis that combining several fine-tuning strategies for gently reducing molecular hydrophobicity might succeed in converting the bactericidal pentapeptide to a borderline hydrophobic, more useful analog. 

## 2. Materials and Methods

**Bacteria:***Escherichia coli* strains: 25922 and 35218 (ATCC, Manassas, VA, USA), 14182 (clinical isolate), Ag100 and Ag100A [34] (Δ*acrAB*) are two K-12 isogenic mutantsand the engineered mutant ML-35p [35]. Additional ESKAPE species [36] tested: *Klebsiella pneumoniae* strains 1287 and 224 (clinical isolates), *Acinetobacter baumannii* ATCC strain 19606, *Pseudomonas aeruginosa* ATCC strains 9027 and 27853. Bacteria were grown in Luria–Bertani (LB) broth (0.5% NaCl, 0.5% yeast extract, 1% tryptone, pH = 7), except for *E. coli* ML-35p that was grown in tryptic-soy broth. Note that LB was used for comparison purposes with previous OAC publications, and that replacing LB with cation adjusted Mueller Hinton broth resulted in essentially identical outcomes [32].

**Peptides:** Unless otherwise stated, all peptides were produced in house by the solid phase method using 9-fluorenylmethoxycarbonyl active ester chemistry on rink amide 4-Methylbenzhydrylamine resin (100–200 mesh, Iris Biotech, Germany), as described [37]. 

Peptide self-assembly was determined by static light scattering measurements as described [32]. Hemolysis was determined by measuring hemoglobin leakage from washed human RBCs, as described [31]. Minimal inhibitory concentration (MIC) was determined using the microdilution assay, as described [32]. Bactericidal kinetics were determined by mixing bacteria with rifampin, OAC, or their combinations, as described [38]. Data were obtained from three independent assays performed in duplicate. 

**Bacterial sensitization:** Sensitization to antibiotics was determined using the checkerboard method in presence of sub-MIC OAC (2.5, 5, and 10 μM), as described [31]. The synergistic effect of the combinations was expressed in terms of the Sensitization Factor (SF), where SF = (MIC antibiotic alone)/(MIC antibiotic upon combination). Data were obtained from three independent assays performed in duplicate. 

Sensitization to plasma components was assessed by mixing bacteria with serial two-fold OAC dilutions in 80% human plasma (Israel Blood Bank) or plasma from the specified species (Technion preclinical research authority or VetSource), as described [29]. Data were obtained from three independent experiments. 

**Outer Membrane damages:** OM permeabilization was investigated using the OM impermeable hydrophobic fluorescent dye 1-*N*-phenylnapthylamine (NPN), as described [39]. Data were obtained from three independent experiments performed in triplicate. For maximal fluorescence, 10 μM PMB [40,41] were used. 

**Dansyl-polymyxin displacement assay:** Commercial PMB sulfate (Sigma P4119) was covalently attached to dansyl chloride and assessed as described [42]. Mono-dansyl Polymyxin B (DPMB) was purified by RP-HPLC. Next, 180 μL of 5 mM HEPES containing 3 μg/mL LPS (from *E. coli* or *P. aeruginosa*) and 2 μM mono-DPMB were incubated in a 96-well plate with 20 μL of the tested compound for 1.5 h at room temperature and fluorescence (excitation: 340 nm, emission: 485 nm) was measured immediately (Synergy HT, BioTek Instruments, Winooski, VT, USA). 

**Inner Membrane Damages:** Damage inflicted to the cytoplasmic membrane was assessed using 3,3-dipropylthiadicarbocyanine iodide (DiSC_3_(5)), a lipophilic potentiometric dye that changes its fluorescence intensity in response to changes in transmembrane potential. Bacteria were grown overnight, diluted and at mid-log were adjusted to O.D = 0.1 (600 nm), centrifuged (10,000 RCF, 5 min), and re-suspended in the assay buffer (5 mM HEPES containing 20 mM glucose, 0.2 mM EDTA, and 50 mM KCl). Then, DiSC_3_(5) dye was added (to final concentration 4 μM) and incubated at 37 °C for 60 min in the dark to allow dye uptake. An aliquot (180 μL) of the bacterial suspension was placed in a 96-well plate and fluorescence was monitored until baseline stabilization (excitation, 622 nm; emission, 670 nm, monitored using Synergy HT, BioTek Instruments, Winooski, VT, USA). A solution (20 μL) containing OAC was added to obtain the desired final concentration. Fluorescence was immediately monitored continuously for 30 min. Reported results are from three independent experiments.

**Intracellular ATP levels** of *E. coli* 25922 (1.5 × 10^8^ CFU/mL) were determined 1 h after incubation with or without OACs using commercial Luciferase-based bioluminescence Assay Kit HSII (Roche diagnostics GmbH, Mannheim, Germany), according to the manufacturer’s instructions. 

**Animals:** All animal studies were performed using male ICR mice (25 ± 2 gr). The Technion Animal Care and Use committee approved all procedures, care, and handling of animals. Ethics approval codes: IL0800519, IL0640421, IL0550618, IL1811217.

The maximum tolerated dose (MTD) was determined after a single-dose subcutaneous (S.C) administration of C_14(ω5)_OOc_10_O using three mice per dose. Animals were monitored for adverse effects for 7 days after injection. 

For efficacy assessments, three infection models were used including one with topical treatment and two with systemic treatment.
Excisional skin wound infection model: mice were anesthetized by intraperitoneal administration of a mixture of ketamine 100 mg/kg and xylazine 5 mg/kg in PBS and their backs shaved with electric clippers. The following day mice were similarly anesthetized and were administrated (S.C) 0.05 mg/kg buprenorphine (for pain relief). A 5 mm diameter piece of skin was removed from the middle of the shaved back, with sterile biopsy punch resulting in a full-thickness injury. A total of 20 µL of a mid-logarithmic culture, containing 5 × 10^6^ CFU *P. aeruginosa* 27853 were applied on the wound. Then, 15 min after inoculation, about 50 µL of hypromellose gel (prepared as described [43]) containing OAC, antibiotic, or their combination were applied on the skin and covered with a piece of medical tape. As a control, the vehicle (drug-free gel) was similarly applied on the skin. After a treatment period of 4 h, about 8 mm diameter of skin surrounding the infected area was removed, suspended in PBS, homogenized, serially diluted 10-fold, and plated for CFU enumeration. The number of viable bacteria was determined after overnight incubation at 37 °C. The lower limit of detection was 50 CFU/wound.Thigh infection model (TI): mice were inoculated intramuscularly with 1 × 10^6^ CFU/mouse of mid-logarithmic *E. coli* 25922 and treated subcutaneously 1 and 3 h after inoculation. Mice were sacrificed 24 h after infection, their thighs excised, homogenized, serially diluted 10-fold, and plated for CFU enumeration. The number of viable bacteria was determined after overnight incubation at 37 °C. The lower limit of detection was 50 CFU/thigh.Urinary tract infection model (UTI): mice were anesthetized via intraperitoneal injection of 100 mg/kg ketamine and 5 mg/kg xylazine. Mice penises were lubricated with an analgesic 2% lidocaine gel. Then, mice were infected with 50 µL of 1 × 10^8^ CFU/mouse of *E. coli* UPEC CFT073, administrated by an intra-urethral injection using a catheter (24 GA, 0.156 IN, 0.7 × 14 mm). Mice were subcutaneously treated with OAC at 1 and 6 h post infection.Mice were sacrificed 24 h post inoculation, the bladder and kidneys were excised, homogenized, serially diluted 10-fold, and plated for CFU enumeration. The number of viable bacteria was determined after overnight incubation at 37 °C. The lower limit of detection was 50 CFU/organ.

### Blood Circulating Concentrations and Organ Bio-Distribution of C_14(ω5)_OOc_10_O

C_14(ω5)_OOc_10_O was subjected to preliminary pharmacokinetics (PK) analysis to determine its plasma concentrations or organ bio-distribution following S.C administration to non-neutropenic pathogen-free mice. OAC quantification was performed by LC-MS as follows: blood was drawn at various time intervals and plasma was separated by centrifugation (5000 RCF, 10 min). Organs were excised and homogenized. Then, OAC was extracted from both plasma and organ homogenates by incubation with 50% acetonitrile: 50% methanol at room temperature with shaking for 30 min, and subsequent centrifugation (5000 RCF, 10 min). Supernatants were diluted two-fold in distilled water and analyzed by LC-MS (ULC ultimate 3000 DIONEX, MS BRUKER Maxis impact). Quantification was based on standard calibration curves, prepared by spiking fresh mice plasma (for PK) or water (for organ bio-distribution) with various amounts of the tested OAC (final concentrations from 2.5 to 50 µg/mL) and subjected to an identical procedure as described above.

**Statistics:***p*-values were calculated using a 1-tailed *t* test (assuming unequal variance). A *p*-value of <0.05 is considered statistically significant.

## 3. Results and Discussion

### 3.1. Derivatives Design and In Vitro Assessment

Table 1 outlines relevant biophysical properties of the reference sequence C_14_KKc_12_K [32] and those of six new derivatives. The first new derivative (C_14_OOc_12_O) represents a sequence alteration involving three lysine-to-ornithine substitutions, presumed to lead to some reduction in molecular hydrophobicity due to ornithine’s shorter side chain [29]. However, these substitutions appear not to lead to biophysical changes substantial enough to be detected by the assays used, including HPLC elution time and bactericidal activity (Figure 2). In contrast, the next alteration (C_14_OOc_10_O) which involved an additional dodecanoyl-to-decanoyl substitution of the A_2_ position, has affected each one of the tested properties. Namely, molecular hydrophobicity was reduced whereas self assembly and hemolysis were deferred to higher concentrations. Moreover, the growth inhibitory concentration in LB has risen (average MIC increased from 3–6 to 12–25 µM, as determined against 12 GNB strains selected from four different species). This was considered a pivotal step towards the desired conversion, as corroborated by the fact that C_14_OOc_10_O also induced a weak but significant antibacterial activity in plasma, namely an activity that was absent when C_14_KKc_12_K or C_14_OOc_12_O were similarly assessed. These effects have further intensified upon additionally substituting the A_1_ position with its unsaturated acyl derivative, yielding C_14(ω5)_OOc_10_O (Figure 1b). In fact, this alteration succeeded in reducing both aggregation and hemolysis to more significant extents (CAC and HC_50_ values became superior to the highest tested concentration, i.e., >100 µM each). Concomitantly, the antibacterial activity in LB was reduced (MIC became >50 µM), yet C_14(ω5)_OOc_10_O has induced a concentration-dependent bacterial killing effect in human plasma (Figure 3a) where the CFU counts were reduced to below the limit of detection at 5 or 10 µM (depending on the plasma donor). This effect revealed to correspond to a bactericidal activity (defined as the ability to reduce the CFU count by three orders of magnitude within 3 h) as evident from time-kill experiments (Figure 3b). Antibacterial activity was not specific to human plasma, as evidenced after a brief survey of diverse animal species (Figure 3 panels c to f). Thus, plasma from human and pig origins appear to be similarly affected by C_14(ω5)_OOc_10_O (Figure 3a,c, respectively) displaying higher antibacterial activity. Note, however, that even though cat and dog plasma samples exhibited high MIC values (≥20 µM) they have nonetheless significantly inhibited proliferation of bacterial inoculums at sub-minimal inhibitory concentrations (Figure 3d,f). 

Additional efforts to reduce hydrophobicity by substituting the A_2_ position with less hydrophobic acyls led to no apparent improvement. Rather, C_14(ω5)_OOc_8_O and C_14(ω5)_OOc_6_O displayed a gradually weaker capacity to potentiate plasma (and antibiotics, as shown in Figure 4), suggesting that acyl length bridging between the cationic residues represents a critical parameter in defining the lipopeptide’s interaction with target bacteria and that it might optimally correspond to a c_8_–c_10_ aminoacyl. Finally, deleting the A_1_ position yielded a significantly more hydrophilic and virtually inactive analog (OOc_12_O) supporting the notion that the interactions leading to potentiation require an optimal hydrophobicity.

Whereas peptide hydrophobicity plays a critical role in the interactions with membranes [44,45,46], our finding that reducing hydrophobicity has increased the plasma antibacterial activity may be rather counter intuitive. Yet, it is reminiscent of reports concerning PMB [47] and related cyclic lipopeptides whose direct antibacterial potency was de-facto substituted for potentiation activity [48] merely by reducing molecular hydrophobicity (e.g., by deleting the fatty acid tail). Furthermore, the overall properties itemized in Table 1 were also reminiscent of previous findings using an OAC analog C_10_OOc_12_O [29,30] proposed to elicit improved activity of innate antibacterial proteins, allegedly through increasing OM permeability. Thus, to corroborate the possibility that similar principles might prevail for C_14(ω5)_OOc_10_O, this analog was similarly tested. Based on the NPN uptake assay (Figure 5a) both C_14_OOc_12_O and C_14(ω5)_OOc_10_O were able to permeabilize the *E. coli* OM but the unsaturated analog appeared more efficient and practically as efficient as PMB (*p* > 0.05) often considered as the “gold standard” for LPS-sequestering agents [49]. Figure 5b provides evidence for the notion that the NPN permeative capacities in presence of C_14(ω5)_OOc_10_O and PMB were significantly hampered by Mg^2+^ thereby suggesting a common (or similar) mode of interaction with the OM. Yet, the fact that PMB was less affected (i.e., the curves in presence and in absence of Mg^2+^ are quite similar for PMB as opposed to the greater change observed for the OAC), could argue for its higher affinity for the cation’s binding sites. This view is supported by additional experiments comparing their abilities to displace binding of dansylated PMB to LPS from *E. coli* (Figure 5c) or from *P. aeruginosa* (Figure 5d), thereby advocating that C_14__(ω5)_OOc_10_O is nearly as efficient as PMB in OM permeabilization despite its lower LPS binding affinity. Note that the binding issue will be further elaborated below, in the proposed mechanism of action.

### 3.2. C_14(ω5)_OOc_10_O Is a Remarkable Antibiotics Potentiator against GNB 

Figure 4 shows the antibiotic’s MICs evolution in absence versus in presence of an adjuvant (C_14(ω5)_OOc_10_O and analogs) at a specified sub-MIC concentration as assessed for rifampin and erythromycin against four GNB species. Figure 4 (left-most upper panel) indicates that while the concentration-dependent trends exhibited some interspecies differences, C_14(ω5)_OOc_10_O was nonetheless able to potentiate rifampin’s action against all four bacterial species, reducing the MIC against *E. coli* and *P. aeruginosa*, from 8 and 32 μg/mL to 0.25 and 1 ng/mL, respectively (i.e., at 10 μM C_14(ω5)_OOc_10_O, rifampin’s MIC were reduced by 32,000 fold for both species). Similarly, rifampin’s MIC against *K. pneumoniae* and *A. baumannii* were both reduced from 32 and 2 μg/mL, respectively, to 0.5 ng/mL. Remarkably, C_14(ω5)_OOc_10_O has reduced rifampin’s MIC values against all four GNB species to values well below the susceptibility breakpoint of *staphylococcus* species (i.e., 1 µg/mL, according to the Clinical Standards Institute) [50]. Noteworthy also is the fact that the lipopeptide’s capacity to reduce the antibiotic’s MIC was hydrophobicity dependent (compare left panel to middle and right panels), thereby correlating rifampin potentiation against GNB and their OM permeabilization (evidenced for *E. coli* in Figure 5). These data again strengthen the notion of a possible role (yet to be determined) for acyl bridge length in rifampin’s permeation and, moreover, highlight a possible causative parallelism between potentiation of antibiotics and potentiation of plasma antimicrobial constituents.

Table 2 shows the sensitization factor (SF) values of two additional published rifampin potentiators, as compared at a single concentration (8 µg/mL each, i.e., 10 µM for C_14(ω5)_OOc_10_O). C_14(ω5)_OOc_10_O was often more potent than the most effective potentiator OAC published so far, i.e., C_10_OOc_12_O [29]. C_14(ω5)_OOc_10_O was also more potent than the PMB derivative SPR741 [48,51] (Figure 1c). Combined, these data suggest that flexible smaller compounds may be more advantageous for efficient antimicrobials potentiation against GNB. Possibly, the OAC’s relatively lower LPS binding affinity (Figure 5) could play a facilitating role as such compounds would be less restrained from engaging in additional interactions, for example. 

### 3.3. Mechanistic Studies

To gain insight into the specific role of each protagonist in the synergistic pair, we determined the survival kinetics under synergistic conditions (i.e., bacteria were exposed to solutions composed of 10 μM C_14(ω5)_OOc_10_O or/and 4 ng/mL rifampin) as summarized in Figure 6. The data suggest some interspecies fluctuations in terms of relative effect(s) exerted by each compound on each bacterial species. However, C_14(ω5)_OOc_10_O and rifampin were individually only capable of delaying proliferation (at most), whereas their combination was bactericidal against each of the tested species. Such an outcome sits well with the notion that C_14(ω5)_OOc_10_O merely facilitates rifampin’s inherent bactericidal mode of action by increasing its cytoplasmic accumulation. A similar view was proposed for C_10_OOc_12_O [29] and polymyxin analogs [20,47,48]. The individual time-kill curves obtained with different GNB species may well illustrate this general idea, where Figure 6a, in particular, was key to our interpretation of the survival kinetics, as follows: upon exposure to rifampin alone, *E. coli* bacteria exhibited a transient static phase that lasted at least 6 h before eventually fading out, reaching normal growth levels after 24 h. Figure 6a also indicates that in absence of rifampin, C_14(ω5)_OOc_10_O too has weakly inhibited bacterial proliferation, unlike its saturated analog that was responsible for rapid bacterial death at this concentration range (Figure 2) allegedly due to abrupt IM disruption (Figure 7). Conceivably, therefore, this lack of drastic IM damage in itself raises the possibility that C_14(ω5)_OOc_10_O exerts a similar but weaker damage as reported for equivalent lipophilic compounds that mildly affect IM functions (such as delocalization of membrane proteins [14,52], partial respiration inhibition [53], and/or dissipation of the transmembrane potential [15,54]). Such damages were proposed for various borderline hydrophobic membrane-active compounds found to have temporarily halted proliferation [12] and thus prompted us to monitor the lipopeptide’s effect on the transmembrane potential. For lack of available direct methods, we used the transmembrane potential sensitive dye, DiSC_3_(5) considering the fluorescent signal released in presence of the bactericidal OAC C_12_K-7α_8_ (used as positive control) to reflect lethal depolarization [26]. Indeed, depolarization by the bactericidal analog, C_14_OOc_12_O, displayed a significant dose-response (Figure 7a), whereas the concentration-dependent depolarization obtained at sub-MIC values of C_14(ω5)_OOc_10_O supports the notion that even at the high concentration of 10 μM, only partial depolarization was produced, thereby reinforcing its borderline hydrophobic status.

Furthermore, bacterial ATP concentrations provided additional evidence that support the occurrence of such mild IM damages (Figure 7b) as both the bactericidal and adjuvant analogs (C_14_OOc_12_O and C_14(ω5)_OOc_10_O, respectively) have reduced the intracellular ATP content but the unsaturated analog was less potent, consistently exhibiting significantly lower ATP levels. We submit that lower ATP content could represent a direct consequence of depolarization and perhaps even reflect its extent, for instance if the periplasmic protons required for ATP production [55] leak back into the cytoplasm through cracks allegedly produced by lipopeptide–IM interaction, as proposed for respiration decoupling agents [56]. 

Another line of supporting evidence emerged by inquiring expected consequences of these alleged damages. Since both protons and ATP are required to fuel efflux pumps [57] actions, their reduced concentrations might decrease drug expulsion, which could translate into increased potency of efflux substrates. This hypothesis is supported by the fact that LL-37 (a host defense peptide) [58] or erythromycin [59] (a macrolide antibiotic), are considered as established efflux substrates of the resistance-nodulation-division (RND) superfamily of pump proteins. Indeed, both are inefficient against GNB (Table 3), yet, both have exhibited higher potency against the efflux-deficient mutant strain Ag100A [60]. Under these conditions, C_14(ω5)_OOc_10_O (but not C_14_OOc_12_O) also behaved as an efflux substrate. Surprisingly (despite its efflux) C_14(ω5)_OOc_10_O managed to overcome resilience to erythromycin in all four GNB species, including in extremely resistant strains whose MIC was >512 µg/mL (lower panels of Figure 4). The fact that *Pseudomonas* bacteria were least sensitized may be precisely linked to their propensity to counteract the action of toxic drugs via overexpression of efflux pumps [57]. In addition, it is worth noting the striking resemblance between erythromycin’s potentiation trend and that of rifampin (compare upper and lower panels of Figure 4) as it argues that OM permeabilization could be implicated in potentiation of both drugs.

The overall findings can be assembled to draw a plausible two-stage scenario attempting to explain how C_14(ω5)_OOc_10_O might potentiate the action of such diverse antibacterial substances against GNB (Figure 8). The first stage follows the path described for many cationic host defense peptides, e.g., PMB [61]. Accordingly, bactericidal and potentiator analogs would readily cross the OM, namely as portrayed by the self-promoted uptake hypothesis [62] where the peptide’s bulkier molecular volume causes the OM permeabilization by forcing local reorganization of peptide–lipid A complexes into unstable mixtures that facilitate OM crossing and periplasm invasion. Once in the periplasm, however, these lipopeptide analogs display drastically divergent behaviors: C_14_OOc_12_O may imbed deeply within the IM thereby inducing its disruption and rapid death, as observed experimentally (Figure 2). Such an outcome is less likely with C_14(ω5)_OOc_10_O for two tightly linked reasons: binding affinity and efflux pumps. Indeed, less hydrophobic analogs typically display a lower membrane-binding affinity [33,63]. As a result, they are more likely to be expelled due to their lingering in the aqueous phase, instead of building up high membrane-bound concentrations leading to lethal membrane perturbations. Data shown in Table 3 argue that C_14(ω5)_OOc_10_O is an efflux substrate, unlike C_14_OOc_12_O. Thus, if part of C_14(ω5)_OOc_10_O molecules manage only a superficial integration of the IM [33,36,60] they could generate milder membrane perturbations (e.g., proton leaks?), eventually leading to partial dissipation of the transmembrane potential. The ensuing dwindled level of periplasmic protons is likely to affect a broad range of membrane functions including efflux which, in GNB, is often carried out by RND [64] and/or ABC [57] pumps. Collectively, therefore, the data support the view that GNB sensitization to erythromycin could be a consequence of lower bacterial respiration and ATP chemiosmosis (Figure 7). We propose that bacterial sensitization to animal plasma could be explained by these or similar considerations, i.e., if plasma resistance of the tested bacteria is related to low permeability of host defense proteins and peptides and/or to their efflux [16], both problems would be addressed by C_14(ω5)_OOc_10_O, as reported herein. Alternatively, plasma resistance may be linked to bacterial virulence factors (e.g., pseudomonal alkaline protease which cleaves C2 complements, thereby blocking both classical and lectin pathways) [65]. In this case too, C_14(ω5)_OOc_10_O might overcome the problem, as suggested by investigations of analogous borderline hydrophobic OACs linking partial depolarization of staphylococci to inhibition of virulence and resistance factors [18,19].

The wild-type *E. coli* strain AG100 and its isogenic Δ*acrAB* mutant AG100A were used to determine the MIC of OACs and of two known acrAB-TolC substrates: the AMP LL-37 and the macrolide antibiotic erythromycin that are normally inefficient against GNB.

### 3.4. In Vivo Studies

To evaluate the potential for therapeutic applications we performed preliminary toxicity, biodistribution, and efficacy experiments using a variety of mouse models. First, we tested the excisional skin wound infection model to assess the effect of topical treatment of *P. aeruginosa*, which was selected for its clinical importance and staggering ability to colonize skin wounds. As shown in Figure 9, the vehicle-treated control experiment enabled some increase in CFU count (i.e., displayed an average difference with initial inoculum of +0.2 log_10_ CFU) whereas application of high concentrations of rifampin or lipopeptide revealed a biocidal capacity. Thus, the rifampin containing gel (1%, i.e., 10 mg/mL, representing standard drug concentration for therapeutic gels and corresponding to MIC × 312) led to a reduction by 1.7 log_10_ units and the C_14(ω5)_OOc_10_O-containing gels (applied at two concentrations: 0.2 and 1%, respectively, corresponding to MIC × 20 and 200, assuming a MIC of 100 µM) succeeded in reducing the CFU counts significantly (i.e., by 0.5 and 1.9 log_10_ units, respectively). Yet, upon combination (i.e., when the wounds were treated with gels containing mixtures of both rifampin and C_14(ω5)_OOc_10_O) the treatments exhibited synergy of action at both concentrations (i.e., CFU counts were reduced by ~4 and >4 orders of magnitude, respectively). The ability to eradicate infecting organisms can be highly desirable, particularly for *P. aeruginosa*, known to render infection resolution very difficult, namely due to weak penetration of antibiotics and host clearance mediators such as antibodies and phagocytes [66,67]. 

Towards assessing C_14(ω5)_OOc_10_O ability to affect infections after systemic treatment, we initially determined the maximal tolerated dose (MTD) following subcutaneous injections to ICR uninfected mice at increasing doses (i.e., 0, 10, 20, 30, and 40 mg/kg/mouse). All mice survived the administered doses after monitoring for 7 days and no signs of toxicity-related stress were visible, arguing for an MTD value higher than 40 mg/kg, which is considerably higher than the highest MTD so far observed with published, systemically active OACs (i.e., typically 20–30 mg/kg) [29].

Guided by these findings, we next assessed the effect of systemic sub-MTD treatments using two *E. coli* mouse infection models: the thigh muscle infection and the urinary tract infection.

In the thigh model, mice were inoculated intramuscularly, treated subcutaneously, and their CFU/thigh enumerated 24 h after infection. In the UTI model, mice were infected by an intra-urethral injection, treated subcutaneously, and their CFU enumerated in bladder and kidneys, 24 h post inoculation. It should be noted that the treatment here involved only C_14(ω5)_OOc_10_O, in order to verify its capability to sensitize GNB to plasma bactericidal components, which could then result in curbing bacterial infections as observed in combination with the bactericidal antibiotic, rifampin (Figure 9).

Figure 10a shows that C_14(ω5)_OOc_10_O was unable to reduce bacterial loads in the thigh model. The same treatment, however, was efficient in the urinary tract infection model (Figure 10b), having reduced the kidneys’ CFU counts (by up to >2 orders of magnitude) but not those of the bladder. Note that in the PBS-treated control experiment two infected mice died at ~14 h post-infection and hence, their CFU counts were not taken into consideration (if they were, then the gap with the treated group would be of four orders of magnitude). 

To shed some light into these seemingly conflicting in vivo findings, we attempted to evaluate the lipopeptide’s biodistribution. As shown in Figure 10c, its plasma concentrations (as determined by quantitative liquid chromatography–mass spectrometry following subcutaneous administration) argue that intact C_14(ω5)_OOc_10_O has rapidly achieved circulating levels of magnitudes comparable to those of classical antibiotics [68] and C_10_OOc_12_O [29]. Thus, the lipopeptide’s maximal plasma concentration (8.7 ± 2.3 µM) was reached approximately after 1 h from administration but potentiating concentrations (i.e., >2.5 µM) were maintained for at least 3 h thereafter. Based on these findings, we next estimated the lipopeptide’s organ biodistribution, targeting seven additional tissues of interest. Values presented in Figure 10d suggest that 3 h post administration, the lipopeptide preferentially accumulated in liver and kidneys at concentrations averaging at least 3 times higher than in the lymph nodes, thighs, bladder, brain, or spleen. To some extent, therefore, these findings tend to align well with those observed in Figure 10a,b and might explain them in the sense that the treatments efficacies correlate well with the adequate lipopeptide’s presence. Beyond findings evidently pertaining to the efficacy outcomes, the lipopeptide’s rapid passage in the lymph nodes and conversely, its accumulation in the liver (Figure 10d), are noteworthy for suggesting merits of future follow-on studies investigating implication on the antibacterial efficacy reported herein, as well as the potential toxicity that might emerge from excessive C_14(ω5)_OOc_10_O accumulation and/or its potential efficacy in affecting hepatic infections and tumors [69]. 

## 4. Conclusions

The presented data suggest that C_14(ω5)_OOc_10_O affects permeability of GNB similarly to reported OACs and polymyxins, albeit exhibiting lesser LPS-binding affinities but significantly higher potentiation capacities. Therefore, C_14(ω5)_OOc_10_O represents a new member of a growing family of linear miniature lipopeptides practically devoid of direct antibiotic activity at circulating concentrations but having the ability to facilitate the tasks of diverse and perhaps more appealing antibacterial substances as postulated in our previous study [29] of the analog, C_10_OOc_12_O where we showed that the OAC induced antibacterial activity in human plasma, which was antagonized by heat treatment, suggesting the proteinaceous nature of the antibacterial factors. This was supported by the synergistic effect observed between C_10_OOc_12_O and exogenous lysozyme in broth and serum media. Moreover, this activity was suppressible by anti-complement antibodies, pointing to the possibility that the lipo-peptides permeabilize GNB to plasma complements as observed with PMBN that sensitized *E. coli* and *S. typhimurium* to the bactericidal complements [20,23]. In fact, beyond antibiotics potentiation, sensitization of pathogenic GNB to antibacterial innate immune mechanisms could endow infected animals with the possibility to benefit from more holistic strategies for resolving infections while minimizing the risk of pro-inflammatory complications that might be associated with biocidal drugs. 

## Figures and Tables

**Figure 1 pharmaceutics-13-01947-f001:**
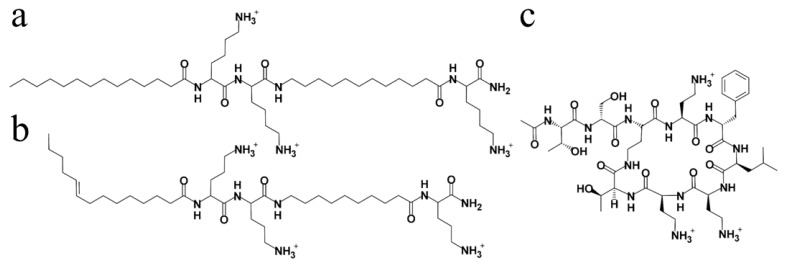
Molecular structures of two main tested OACs and a comparator. (**a**) C_14_KKc_12_K (MW: 809); (**b**) C_14(ω5)_OOc_10_O (MW: 737); (**c**) PMB derivative SPR741 (MW: 992). In (**a**,**b**) the C-terminus is amidated, the letters C and c, respectively, denote an acyl and aminoacyl whose length (number of carbon atoms) is defined by the subscript; the parenthesis _(ω5)_ in (**b**) denotes the position of a double bond; K and O represent the amino acids lysine and ornithine, respectively.

**Figure 2 pharmaceutics-13-01947-f002:**
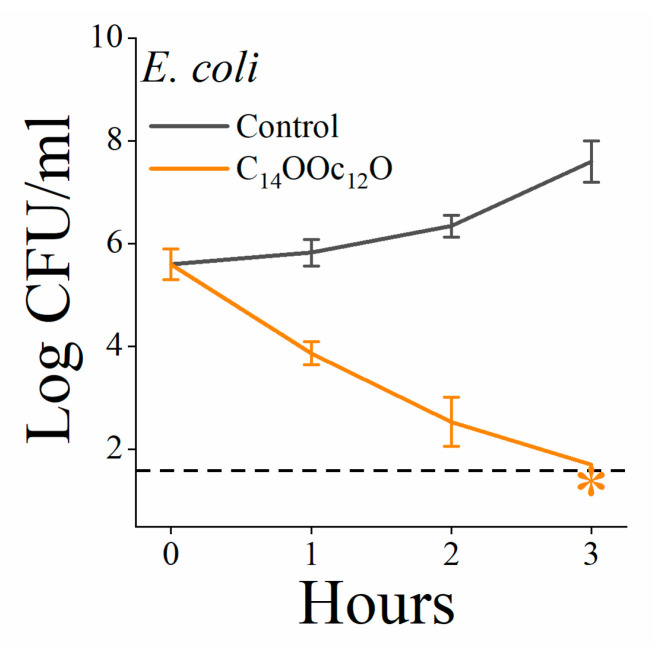
Time-kill kinetics. The bactericidal activity of 10 µM C_14_OOc_12_O (orange trace) is demonstrated in comparison to the untreated control (black trace) in LB. The dashed horizontal line represents the limit of detection (log_10_ 50 CFU/mL = 1.69). Orange asterisk (*) denotes lack of detectable CFU.

**Figure 3 pharmaceutics-13-01947-f003:**
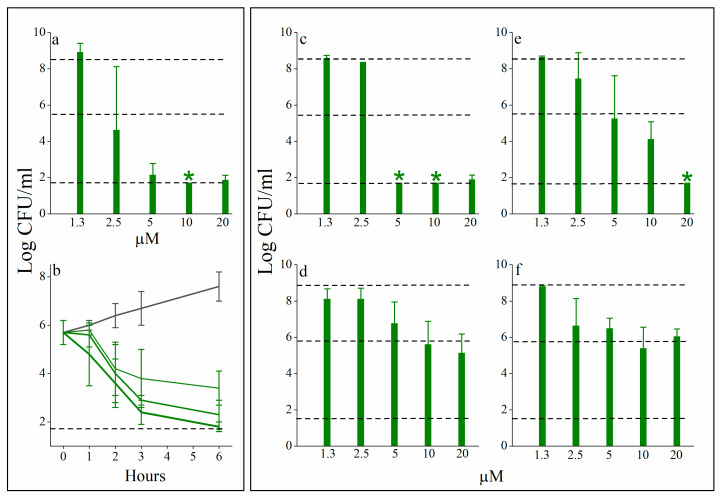
Animal plasma antibacterial activities against *E. coli* 25922. (**a**) Dose-dependent reduction of colony forming units (CFU) as determined 24 h after exposure to C_14(ω5)_OOc_10_O in human plasma (80% final plasma concentration in PBS). Values represent the mean ± SD obtained from three individual plasma donors (mean inoculum was 1.6 ± 0.8 × 10^6^ CFU/mL). (**b**) Time-kill kinetics determined upon *E. coli* exposure to 0, 2.5, 5, and 10 µM C_14(ω5)_OOc_10_O (from top to bottom, respectively). Values represent the mean ± SD obtained from two individual plasma samples, each subjected to two independent assays. Panels (**c**–**f**), respectively, show the OAC’s concentration-dependent activities in plasma samples obtained from pig, cat, sheep, and dog. The three dashed horizontal lines respectively represent the average CFU count of the vehicle control (upper line), the inoculum (middle line), and the limit of detection (lower line), i.e., log_10_ 50 CFU/mL = 1.69. Green stars (*) denote lack of detectable CFU.

**Figure 4 pharmaceutics-13-01947-f004:**
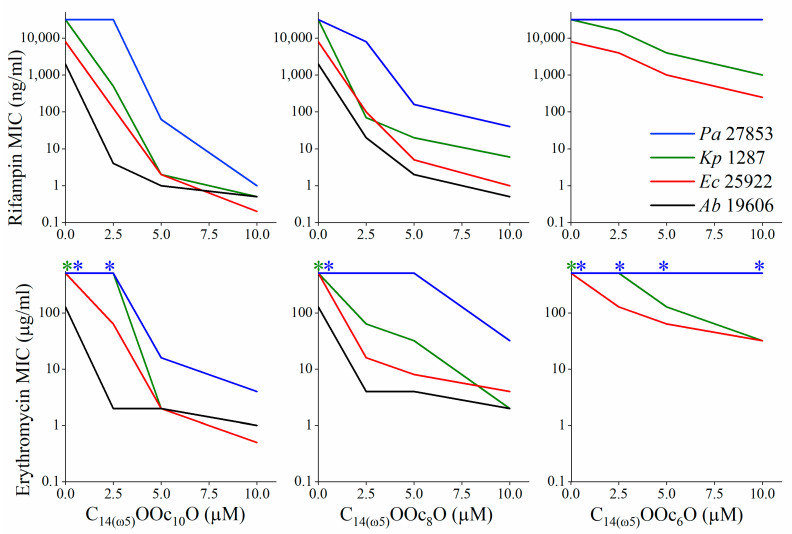
Hydrophobicity dependence of antibiotics potentiation by lipopeptide analogs. Sensitization of GNB to rifampin and erythromycin (upper and lower panel, respectively) was assessed by MIC experiments performed in LB medium, in presence or absence of an OAC analog whose A_2_ position was gradually shortened (from c_10_ to c_6_). *Pseudomonas aeruginosa* 27853 (*Pa*, blue), *Klebsiella pneumoniae* 1287 (*Kp*, green), *Escherichia coli* 25922 (*Ec*, red), and *Acinetobacter baumannii* 19606 (Ab, black). Asterisks (*) denote MIC values greater than the highest tested concentration (i.e., >512 µg/mL).

**Figure 5 pharmaceutics-13-01947-f005:**
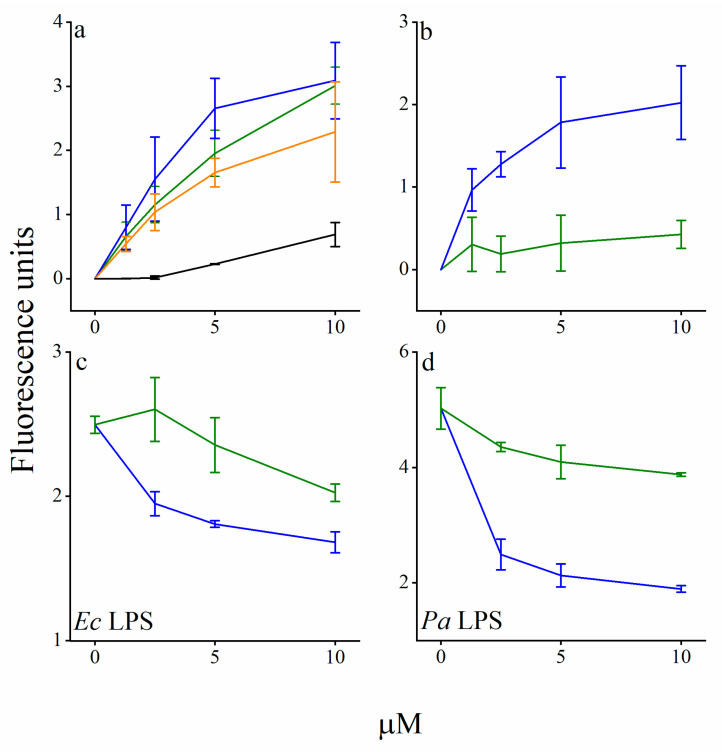
Lipopeptide capacities to affect *E. coli* outer membrane permeability. (**a**) Outer membrane (OM) permeabilization to the hydrophobic dye NPN was determined 10 min after bacteria (*E. coli* 25922, 2 × 10^8^ CFU/mL) were exposed to each peptide (5 µM) in NPN-containing HEPES at 37 °C. *p* << 0.05 for comparing C_14_OOc_12_O to C_14(ω5)_OOc_10_O or to PMB, and *p* > 0.05 for comparing C_14(ω5)_OOc_10_O to PMB. Color code (panels (**a**–**d**)): green, C_14(ω5)_OOc_10_O; orange, C_14_OOc_12_O; black, OOc_12_O; blue, polymyxin B (PMB). (**b**) OM permeabilization (as in panel **a**) in presence of 10 mM MgCl_2_; (**c**,**d**), Dansyl-PMB displacement assay using LPS from *Escherichia coli* and *Pseudomonas aeruginosa*, respectively, as measured 1.5 h after incubation in HEPES with C_14(ω5)_OOc_10_O (green) or PMB (blue).

**Figure 6 pharmaceutics-13-01947-f006:**
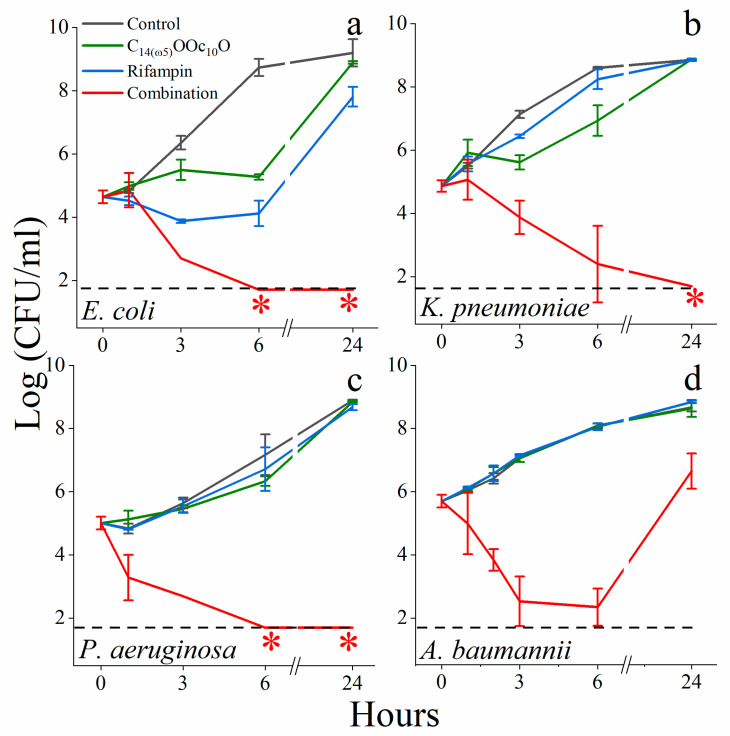
Time-kill of selected ESKAPE bacteria.Bacteria were cultured in LB medium in absence of a drug (black traces) or in presence of 10 μM C_14(ω5)_OOc_10_O (green traces), 4 ng/mL rifampin (blue traces), or their combination (red traces). Error bars represent standard deviations. The dashed horizontal line represents the limit of detection (log_10_ 50 CFU/mL = 1.69). Red asterisks denote lack of detectable CFU.

**Figure 7 pharmaceutics-13-01947-f007:**
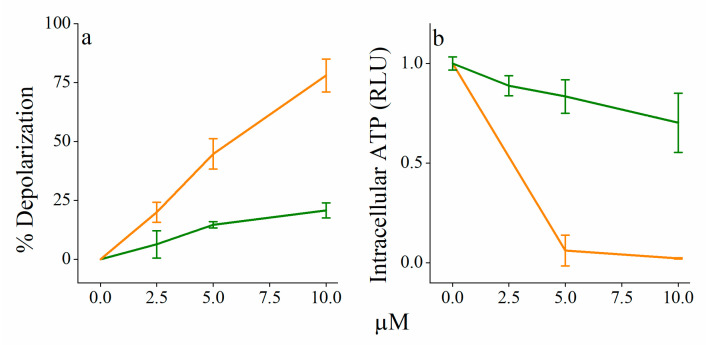
Evidence for proton and ATP leakage across the inner membrane. (**a**) Dissipation of the transmembrane potential in *E. coli* 25922 (8.8 ± 1.8 × 10^7^ CFU/mL) pre-incubated with DiSC_3_(5) as determined 15 min after exposure to C_14_OOc_12_O (orange) or to C_14(ω5)_OOc_10_O (green). Data represent percent depolarization as compared to the positive control, 50 µM C_12_K-7α_8_ [10]. (**b**) Intracellular ATP concentrations were determined after 1 h incubation with C_14(ω5)_OOc_10_O (green) or C_14_OOc_12_O (orange) using *E. coli* 25922 at 5 × 10^7^ CFU/mL. RLU, relative luminescence units.

**Figure 8 pharmaceutics-13-01947-f008:**
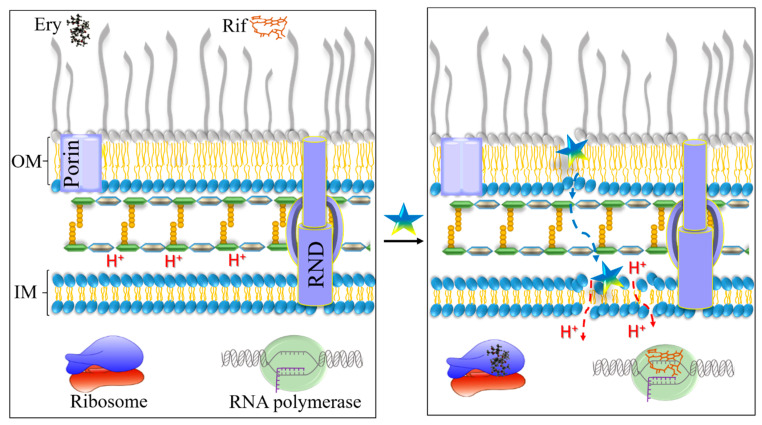
Proposed mechanism for GNB sensitization by facilitating drug interactions with their cytoplasmic targets. The left panel depicts the general organization of the two-membrane system prevailing in GNB cell wall. The outer membrane (OM) and inner membrane (IM) are separated by a peptidoglycan-containing periplasmic space where protons (H^+^) normally accumulate to support the trans-membrane potential. Porin and RND, respectively, represent a passive and an energy-dependent metabolic protein gate. Hydrophobic antibiotics such as rifampin (Rif) and erythromycin (Ery) are depicted floating above the OM layer, to reflect their low permeability, impeding interaction with their cytoplasmic targets (the RNA polymerase and ribosome, respectively). The right panel highlights the reported effects of C_14(ω5)_OOc_10_O (represented by a pentameric star): initially, the lipopeptide destabilizes the OM thereby facilitating OM translocation of itself and that of low permeability antibiotics. Its subsequent superficial interaction with the IM would partially perturb various IM-linked functions such as active transport, hence the observed potentiation of efflux substrates, allegedly resulting from cytoplasmic accumulation, as exemplified by Ery.

**Figure 9 pharmaceutics-13-01947-f009:**
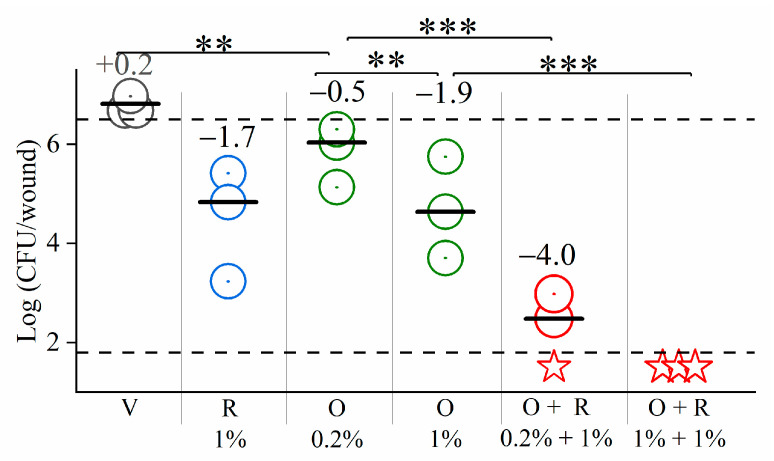
Excisional skin wound infection model. Plotted are colony forming units (CFU) counts of *P. aeruginosa* 27853 in each infected mouse (*n* = 3 per group) upon topical treatment by a drug-containing hypromellose gel. V, vehicle (drug-free hypromellose gel); R, rifampin; O, C_14(ω5)_OOc_10_O. Percentages denote the drugs concentration (*w*/*v*) in the gel. The numbers above each column indicate the change from the initial inoculum (represented by the upper dashed line). Horizontal bars specify the median. The limit of detection is represented by the lower dashed line (log_10_ 50 CFU/mL = 1.69). Statistically significant differences are denoted by double (**) and triple (***) asterisks for *p*-value < 0.05 and *p*-value < 0.005, respectively. Red stars denote lack of detectable CFU.

**Figure 10 pharmaceutics-13-01947-f010:**
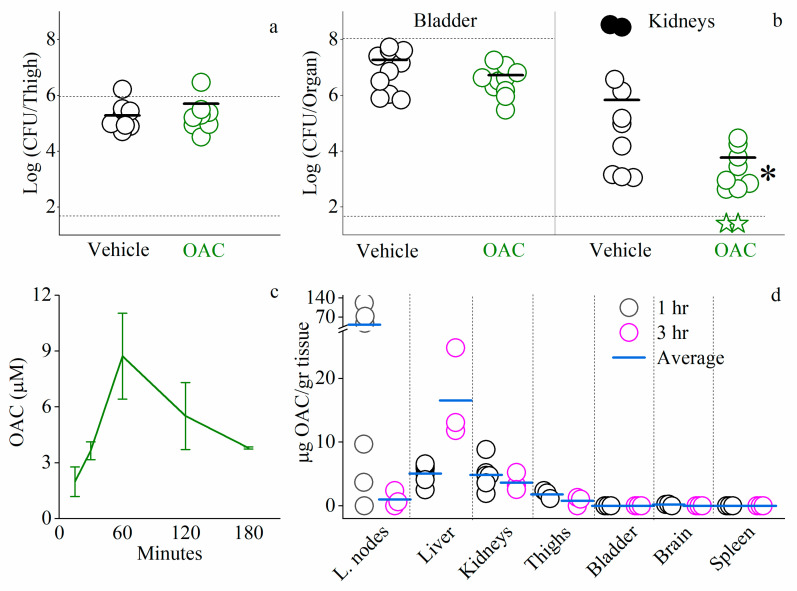
Systemic efficacy studies using mouse infection models. (**a**,**b**) Data points represent colony forming unit (CFU) counts harvested from infected mice (8 and 10 mice per group, respectively). Panel (**a**) depicts results of the thigh infection model where mice were inoculated intramuscularly with *E. coli* 25922 (upper dashed line represents the inoculum) and treated subcutaneously with C_14(ω5)_OOc_10_O (12.5 mg/kg) at 1 and 3 h post-infection. Panel (**b**) depicts results of the urinary tract infection model where mice were infected with *E. coli* UPEC CFT073 by intra-urethral injection and treated subcutaneously with C_14(ω5)_OOc_10_O (12.5 mg/kg) at 1 and 6 h post-infection. Solid circles denote mice that died before the experimental endpoint. Green stars denote lack of detectable CFU. Horizontal bars specify the average. The juxtaposed asterisk (*) denotes statistically significant difference (*p*-value = 0.003 or 0.008, if the dead mice are not included). The limit of detection is represented by the lower dashed line (log_10_ 50 CFU/mL = 1.69). Panels (**c**,**d**) show the evolution of C_14(ω5)_OOc_10_O concentrations in mouse plasma (**c**) or in organs (**d**) after single subcutaneous administration (12.5 mg/kg) to ICR mice (data in c are from two mice/time points except for 60 and 180 that were 8 and 3, respectively) as determined by liquid chromatography–mass spectrometry. Error bars represent standard deviations. The limit of detection was 0.5 ppm (0.68 µM).

**Table 1 pharmaceutics-13-01947-t001:** Biophysical attributes of relevant lipopeptide analogs.

Lipopeptide Sequence	H (%)	CAC (µM)	HC_50_ (µM)	MIC (µM)	
				LB ^a^ Medium	Human ^b^ Plasma
* C_14_KKc_12_K	55	20 ± 5	12 ± 1	3–6	>20
C_14_OOc_12_O	55	15 ± 1	14 ± 4	3–6	>20
C_14_OOc_10_O	53	45 ± 14	28 ± 2	12.5–25	10–20
C_14(ω5)_OOc_10_O	50	>100	>100	>50	2.5–5
C_14(ω5)_OOc_8_O	48	>100	>100	>50	2.5–5
C_14(ω5)_OOc_6_O	47	>100	>100	>50	5
OOc_12_O	24	>100	>100	>50	>20

*, Reference peptide [32], shown for comparison purposes. Grey background specifies published data; H, hydrophobicity, defined as % acetonitrile required for elution in reversed phase HPLC using a C18 column. Values were rounded to nearest whole number; CAC, critical aggregation concentration, determined by light scattering in PBS; HC_50_, lipopeptide concentration that caused 50% hemolysis compared to water (determined by measuring hemoglobin leakage after 3 h incubation in PBS at 37 °C, using 1% washed human erythrocytes); MIC, minimal inhibitory concentration, determined in LB medium and in plasma, using OD measurements and CFU counts, respectively; ^a^, mean of 12 GNB strains, specified in Section 2; ^b^, mean of three donors, assessed on *E. coli* 25922.

**Table 2 pharmaceutics-13-01947-t002:** Sensitization of Gram-negative bacteria to rifampin.

	Sensitization Factor at 8 µg/mL		
Bacteria	C_14(ω5)_OOc_10_O	C_10_OOc_12_O	SPR741
*Kp*	**64,000**	8000 [30]	32 [51]
*Ec*	**32,000**	16,000 [30]	8192 [51]
*Pa*	**32,000**	**1000**	5 [48]
*Ab*	**4000**	**4000**	256 [48]

Comparing C_14(ω5)_OOc_10_O sensitization extents with those of two published adjuvants; sensitization factor is the ratio (rifampin MIC alone)/(rifampin MIC in combination) at the specified adjuvant concentration; Kp, *Klebsiella pneumoniae*; Ec, *Escherichia coli* 25922; Pa, *Pseudomonas aeruginosa* 27853; Ab, *Acinetobacter baumannii* 19606; highlighted in bold fonts are values determined in the present study. Note: SF values of the PMB analog SPR741 were obtained using the same *Ab*, *Pa*, and *Ec* (but not *Kp*) strains.

**Table 3 pharmaceutics-13-01947-t003:** Effect of RND pumps on MIC values of diverse antimicrobials.

Tested Compound	MIC (µM)	
	Ag100	Ag100A
LL-37	22.2	1.1
Erythromycin	174.4	10.9
C_14(ω5)_OOc_10_O	25	6.2
C_14_OOc_12_O	3.1	3.1

## Data Availability

Not applicable.

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
