# Peer review of "Mechanistic Studies of Antibiotic Adjuvants Reducing Kidney’s Bacterial Loads upon Systemic Monotherapy"

_pharmaceutics, 2021, doi:10.3390/pharmaceutics13111947_

Round 1

Reviewer 1 Report

Manuscript entitled “Antibiotic Adjuvant Reduces Kidneys Bacterial Loads of Infected Mice Upon Systemic Monotherapy” by Zaknoon et al. developed pentapeptide-like derivative of the reference sequence C14KKc12K and tested its antibiotic adjuvant efficacy in vitro and in vivo against Gram negative bacteria. Acyl derivative C14(ω5)OOc10O caused enhanced bacterial outer membrane permeability and enhanced transport of antibiotics inside bacteria. Moreover, C14(ω5)OOc10O reduced kidney bacterial load and exhibited synergistic effect with antibiotic rifampin in skin-wound infection model.

This study in well designed and exhibited the adjuvant efficacy of synthetic an acyl derivative against the Gram negative bacteria. The following suggestions should be addressed to further improve the manuscript.

(1) Does outer membrane permeability (Figure 1) significantly enhance? There is no significant value.

(2) Efficacy of C14OOc12O is close to C14(ω5)OOc10O. Why it was not tested in Hydrophobicity-dependence of antibiotics potentiation in Figure 4?

(3) There are some complex sentences in the manuscript that need editing.

(4) Hour should be uniform throughout the manuscript. In some spaces it is written as hr.

Author Response

The authors thank the reviewers for their valuable comments on our manuscript and for helping improve its quality. We have addressed all comments and made the corresponding corrections as detailed below. We hope the revised manuscript meets the requirements for publication.

Reviewer 1:

  1. Does outer membrane permeability (Figure 1) significantly enhance? There is no significant value.

The said “Figure 1” only depicts molecular structures. Therefore, the reviewer must be referring to Figure 3, which shows, indeed, the outer membrane permeability changes caused by OACs as compared to PMB. We concluded that C14(ω5)OOc10O is nearly as potent (i.e., has P values >0.05)  precisely as its permeability curve shows no significant difference with that of the known gold standard, PMB. For the this reason we opt for no change in the manuscript, with all due respect.

  1. Efficacy of C14OOc12O is close to C14(ω5)OOc10 Why it was not tested in Hydrophobicity-dependence of antibiotics potentiation in Figure 4?

We respectfully disagree. We consider that testing the antibiotics potentiation capacity of C14OOc10O is unlikely to provide a valuable information for the advancement of the study goals simply because C14OOc10O would directly kill the bacteria, as shown in Table 1 (MIC = 3 µM) and in Figure 5a (bactericidal kinetics).   

  1. There are some complex sentences in the manuscript that need editing.

We believe to have addressed this comment throughout the revised manuscript according to the suggestions made by the editor and both reviewers.

  1. Hour should be uniform throughout the manuscript.

Done.

Reviewer 2 Report

The manuscript by The manuscript by Mor A. et al describes the activity of antibacterial peptides adjuvants in in vitro and in vivo experiments.

The manuscript deserves interest in the sense that it tackles an important topic which is the need of new weapons against bacteria and reports upon a consistent experimental design.

Major drawbacks

  • The proof of efficacy in animals is a very important part of the manuscript, but the results of the treatment with the rifampicin potentiator don’t show statistical significance, at the two selected doses. Since, this lack is only due to the low number of animals, it is all the more important to be addressed.
  • Thigh infection and urinary infection. The results and the MM do not state clearly the treatment that mice underwent to. In the legend of the figure and in the text only the potentiator OAC is indicated, but in the MM the OAC, antibiotic and combination is stated. It is unclear, then, how the mice were treated. The logical experimental setting has to be the combination treatment, because C14(ω5)OOc10O already showed poor antimicrobial activity, which is in facts wanted, as stated in the results session.
  • The title only refers to the animal treatments, but actually most of the articles is focused on the mode of action of the OAC.
  • A discussion focused on the description and comments of OAC acting as bactericidal either by itself in human plasma or in the presence of rifampin is needed, because results are reported in both cases and have to be aligned.
  • The concept of human plasma killing bacteria has to be introduced.
  • The difference between potentiation and synergy has to be addressed. Moreover, synergy assessment’s gold standard is the checkerboard method and FIC determination, which are not mentioned here. I recommend to either present the FIC data or justify.
  • Arrangement of paragraphs/Figures. The whole manuscripts has to be rearranged in such a way that experiments and relative figures are mentioned in the text in an orderly fashion. As an example, Figure 5 is mentioned at page 4, before Figure 2. In the MM session, as well, the different paragraphs have to follow the order of the experiments exposed in the results.
  • Figures’ appearance. Figures 2, 3, 4, 5, 6 can be improved, the lines of the graphs are very thin and there is no color legend. The color legends would help, particularly in figure 4. Figure 8 is exaggeratedly zoomed in.

Minor drawbacks:

  • Table 1. In the legend add the explication of the abbreviations, ie H, hydrophobity….
  • Figure 2 and related results. TKK is most commonly carried out up to 24h so to observe possible regrowth. Please give explanation or show the results at longer times.
  • Correct the paragraph starting at page 7 line 149, it is unclear to what PMB is compared and also the binding displacement has to be better introduced.
  • Figure 6 ‘s legend lacks b)
  • Page 14 line 271 “are normally inefficient against GNB” is not placed in the correct paragraph.
  • Page 14 line 272 “Ag100A” indicate specie and properties/characteristics.

Author Response

The authors thank the reviewers for their valuable comments on our manuscript and for helping improve its quality. We have addressed all comments and made the corresponding corrections as detailed below. We hope the revised manuscript meets the requirements for publication.

  1. The proof of efficacy in animals is a very important part of the manuscript, but the results of the treatment with the rifampicin potentiator don’t show statistical significance, at the two selected doses. Since, this lack is only due to the low number of animals, it is all the more important to be addressed.

We agree. Significance in the combination experiments of Figure 8 was not shown having considered (erroneously) that the CFU count differences were substantial enough and essentially obvious to all. We now have corrected this lack and show the relevant P values.  

  1. Thigh infection and urinary infection. The results and the MM do not state clearly the treatment that mice underwent to. In the legend of the figure and in the text only the potentiator OAC is indicated, but in the MM the OAC, antibiotic and combination is stated. It is unclear, then, how the mice were treated. The logical experimental setting has to be the combination treatment, because C14(ω5)OOc10O already showed poor antimicrobial activity, which is in facts wanted, as stated in the results session.
    We agree and added a paragraph (lines 544 - 549) to that effect.

  2. The title only refers to the animal treatments, but actually most of the articles is focused on the mode of action of the OAC.

We agree. Accordingly, we have modified the title to reflect the mechanistic studies as well.

  1. A discussion focused on the description and comments of OAC acting as bactericidal either by itself in human plasma or in the presence of rifampin is needed, because results are reported in both cases and have to be aligned.

We agree. The mechanism of action was already amply discussed (lines 376 – 502) based on the current findings. Therefore, we have added a short paragraph to cover the plasma bactericidal effect induced in presence of a previously investigated analogous OAC (lines 599 - 606).

  1. The concept of human plasma killing bacteria has to be introduced.
    The new paragraph (added in response to comment # 4) includes the concept of bactericidal plasma effect. In addition, another paragraph covering the plasma bactericidal effect was added in the Introduction section (lines 50 - 52).

  2. The difference between potentiation and synergy has to be addressed. Moreover, synergy assessment’s gold standard is the checkerboard method and FIC determination, which are not mentioned here. I recommend to either present the FIC data or justify.

Generally, measuring FIC in cases where adjuvants have no measurable MIC poses a challenge for the quantification of adjuvant potency. More relevant is the adjuvant’s concentration that lowers the antibiotic’s MIC in resistant bacteria to a value equal to, or lower than the breakpoint concentration, that is, the concentration of antibiotic that defines whether a strain is resistant or sensitive. Specifically, in our case, FIC was not calculated since the MIC for C14(ω5)OOc10O is very high (> 50mM). Therefore, we agree with the reviewer in pointing out that our text does not distinguish between the terms “potentiation” and “synergy”, however there is no need for such a distinction due to the fact that the observed potentiation level was >4 folds (in fact >30,000 folds) which demonstrates that the potentiation effect is more than additive and indicates a synergistic effect by definition.

  1. Arrangement of paragraphs/Figures. The whole manuscripts has to be rearranged in such a way that experiments and relative figures are mentioned in the text in an orderly fashion. As an example, Figure 5 is mentioned at page 4, before Figure 2. In the MM session, as well, the different paragraphs have to follow the order of the experiments exposed in the results.

Done

  1. Figures’ appearance. Figures 2, 3, 4, 5, 6 can be improved, the lines of the graphs are very thin and there is no color legend. The color legends would help, particularly in figure 4. Figure 8 is exaggeratedly zoomed in.
    All done: Appearance of Figures 2, 3, 4, 5, 6 and 8 was improved accordingly.

Minor drawbacks:

Table 1. In the legend add the explication of the abbreviations, i.e. H, hydrophobity….

Done

Figure 2 and related results. TKK is most commonly carried out up to 24h so to observe possible regrowth. Please give explanation or show the results at longer times.

We disagree. Data for bacterial killing after 24 hours already appears in Figure 2a, along with  the dose-dependent kinetics at shorter time periods of exposure, shown in Figure 2b.

Correct the paragraph starting at page 7 line 149, it is unclear to what PMB is compared and also the binding displacement has to be better introduced.
Agreed. We rephrased the paragraph (lines 293-315) to increase clarity. The issue is also elaborated in the proposed mechanism of action discussion (paragraph starting at line 451).

Figure 6 ‘s legend lacks (b).

Done

Page 14 line 271 “are normally inefficient against GNB” is not placed in the correct paragraph.
We have rephrased that paragraph (line 437-442) for increased clarity.

Page 14 line 272 “Ag100A” indicate specie and properties/characteristics.

A brief description of the mutant properties appears in materials and methods (line 70). We also added a relevant reference.

Round 2

Reviewer 2 Report

All my concerns have been addressed